# Sex and estrous cycle affect experience-dependent plasticity in mouse primary visual cortex

**Rachel W. Schecter**[�], **Cambria M. Jensen**[�], **Jeffrey P. Gavornik**[iD] *

Center for Systems Neuroscience, Biology Department, Boston University, Boston, MA, United States of America

she These authors contributed equally to this work.

* gavornik@bu.edu

**Data Availability Statement:** Data and code is available at: https://gavorniklab.bu.edu/supplemental-materials.html.

**Funding:** This work was funded by grants NIMH R00MH099654 and NEI R01EY030200. Neither

## Abstract

Sex hormones can affect cellular physiology and modulate synaptic plasticity, but it is not always clear whether or how sex-dependent differences identified *in vitro* express themselves as functional dimorphisms in the brain. Historically, most experimental neuroscience has been conducted using only male animals and the literature is largely mute about whether including female mice in will introduce variability due to inherent sex differences or endogenous estrous cycles. Though this is beginning to change following an NIH directive that sex should be included as a factor in vertebrate research, the lack of information raises practical issues around how to design experimental controls and apply existing knowledge to more heterogeneous populations. Various lines of research suggest that visual processing can be affected by sex and estrous cycle stage. For these reasons, we performed a series of *in vivo* electrophysiological experiments to characterize baseline visual function and experience-dependent plasticity in the primary visual cortex (V1) of male and female mice. We find that sex and estrous stage have no statistically significant effect on baseline acuity measurements, but that both sex and estrous stage have can modulate two mechanistically distinct forms of experience dependent cortical plasticity. We also demonstrate that resulting variability can be largely controlled with appropriate normalizations. These findings suggest that V1 plasticity can be used for mechanistic studies focusing on how sex hormones effect experience dependent plasticity in the mammalian cortex.

## Introduction

In May of 2014, the NIH released a directive that sex must be factored into research of vertebrate animals [1]. In support of the NIH statement, Shansky and Woolley advise researchers to "accept [sex differences] as part of a complex physiological background" of each animal [2]. This statement minimizes the impact that variability associated with a heterogeneous "physiological background" could have on neuroscience research, where a variety of factors that can affect brain function–handling, cage mate socialization, etc.–are difficult to measure and control. Unlike other fields of biology, systems-level neuroscience research often lacks valid *ex*

institute took part in study design, data collection, or analysis or paid author salary. The funders had no role in study design, data collection and analysis, decision to publish, or preparation of the manuscript.

**Competing interests:** The authors have declared that no competing interests exist.

*vivo* models that could mitigate these factors. Neuroscientists have been hesitant to include female animals in their research since sex chromosomes and gonadal hormones represent two sources of potentially serious variance. Some have raised concerns that the broad inclusion of female animals in preclinical experiments will not produce desired outcomes but may have unintended consequences of "wasting resources, slowing down research or even provoking a backlash" ds [3, 4]. As a consequence of these considerations, there is a 5:1 bias towards male-only neuroscience studies, the highest in all fields measured [5].

Brain research must overcome this experimental inertia both as a practical matter (i.e. to comply with the NIH directives) and to address substantive critiques of building biological science around male animals alone. A major hurdle is the assumption that the estrous cycle introduces variability in both behavioral and physiological measures of neural function [6]. While several recently published meta-analyses suggest this concern may be overblown in rodents [7–9], there is also good reason to take this concern seriously: chromosomal and hormonal effects do lead to clear differences between females and males that can be seen at multiple levels and can impact a variety of functions including cognitive and emotional responses, learning and memory, and degree of severity in a variety of neurological disorders resulting from injury or pathology [10, 11].

Plasticity experiments are particularly susceptible to hormonal cycle influences due to the variety of receptor types and signaling cascades that can be altered by fluctuating neuromodulator activity. For example, estrogen modulates NMDAR subunit expression in the hippocampus [12, 13], changes spine density [14, 15], and can enhance LTP [16, 17]. While there is scant direct evidence for or against sexual dimorphism in sensory cortices, estrogen has been demonstrated to modulate spine density in the imprecisely defined "sensorimotor cortex" [18] and nitric oxide synthase knockout in primary somatosensory cortex affects experience-dependent plasticity in male but not female animals [19]. Sex differences in V1 have not been directly demonstrated *in vivo*, but there are several reasons to expect they exist. Human studies have shown that various aspects of visual perception correlate with fluctuating estrogen levels over the menstrual cycle, including visual memory and spatiotemporal processing [20–22]. *In vitro* animal work has demonstrated that neuromodulators can fundamentally change the form of LTP/LTD induction curves in V1 neurons (switching, for example, whether a particular stimulation pattern results in LTP or LTD) [23, 24], 7α-Estradiol can promote experience-dependent plasticity in rat V1 [25], and mouse V1 is sensitive to estrogen [26] which plays a role in V1 homeostatic plasticity [27]. These findings highlight the importance of determining whether sex and estrous cycles impact V1 function *in vivo*. One thing to note is that nearly all of the work linking estrogen to plasticity was performed *in vitro* or by artificially administering estrogen to gonadectomized animals. As such, the literature provides essentially no information on the extent to which endogenous sex-based variations exist in sensory cortex or how to control for them if they exist.

For all these reasons, we set out to quantify the impact of sex and endogenous hormone fluctuations caused by the estrous cycle on baseline function and experience-dependent plasticity in the primary visual cortex. We measure no significant difference in visual acuity limits between male and female animals. We find that sex has no significant effect on a form of spatial coding called stimulus-selective response potentiation (SRP) [28, 29], but it does affect spatiotemporal sequence potentiation [30]. We also tracked estrous cycling in a parallel set of experiments conducted exclusively in female mice and determined that estrous stage has no impact on our measurements of physiological function but can modulate plasticity coding both spatial and spatiotemporal information. Our results show that these effects can be effectively controlled for in some circumstances using in-group normalization and suggest mouse V1 as an *in vivo* model system to study how sex hormones affect mechanistically distinct forms of cortical plasticity and learning.

## Materials and methods

### Mice

All procedures involving laboratory animals occurred at Boston University and adhered to the guidelines of the National Institutes of Health and were approved by the Institutional Animal Care and Use Committee at BU, Boston, MA, USA (IACUC PROTO201800679). Mice were housed in groups of 2–5 separated by sex with food and water available *ad libitum* and maintained on a 12-hour light-dark cycle. All animals were C57BL6 WT ordered from Charles River or C57BL6 WT progeny of heterozygous breeding pairs of VGAT-ChR2-EYFP transgenic mice (Jax stock #014548), Thy1-GCaMP6f transgenic mice (Jax stock #024276), or DAT-IRES-CRE transgenic mice (Jax stock #006660), genotyped by Transnetyx using real-time PCR for the EYFP, EGFP, and CRE genes, respectively; only animals which lacked transgene expression and tested positive for the corresponding WT control were used in experiments.

### Estrous staging

For estrus/diestrus grouped experiments: female mice 8 weeks and older were analyzed daily as described in [31]. Animals were encouraged to grip onto the cage lid with their front forepaws and held at the base of the tail with the thumb and forefinger, using the middle and ring fingers loosely flanking the mouse's torso underneath the rib cage. Direct cytology was performed with tissue collected via vaginal lavage of fifteen microliters of sterile PBS, and wet mount slides were examined with phase contrast microscopy. Smears were classified (with reference to [32] as follows: estrus—a predominance of cornified epithelial cells, metestrus—a mix of cornified epithelial and leukocyte cells, diestrus–a predominance of leukocyte cells, and proestrus–a predominance of nucleated epithelial cells (see Fig 2). Once mice had completed at least one cycle through all four stages and were at the height of either estrus (100% cornified cells) or diestrus (100% leukocyte cells) they were included in experimental groups as yoked pairs.

### VEP surgery

Electrode implantation followed the procedures used in previous studies [33, 34]. Mice were first injected with 0.1 mg/kg Buprenex sub-cutaneously to provide analgesia. They were then anesthetized with 1.5–3% isoflurane. The scalp was shaved and cleaned with iodine and 70% ethanol before an incision was made to expose the skull. A steel head post was affixed to the skull anterior to bregma using cyanoacrylate glue. Burr holes (< 0.5 mm) were then drilled in the skull over binocular V1 (3.0 mm lateral of lambda). Tapered tungsten recording electrodes (FHC, Bowdoinham, ME, US), 75 μm in diameter at their widest point, were implanted in each hemisphere 450 μm below the cortical surface to target thalamocortical recipient layer 4. Silver wire (A-M systems, Sequim, WA, US) was placed in the cerebrospinal fluid over prefrontal cortex to serve as an electrical reference. Mice were allowed to recover for at least 48 hours prior to initial head-fixation.

### In vivo electrophysiology

VEP recordings were conducted in awake, head-restrained mice. Prior to recording, mice were habituated to the restraint apparatus *in situ* in front of a gray screen for a 30-minute session on each of two consecutive days. All data was amplified and digitized using the commercially available OmniPlex recording system (Plexon Inc., Dallas TX). Data was acquired at 25 kHz and local field potentials (LFPs) were down-sampled to 1-kHz utilizing a 500-Hz low-pass

anti-aliasing filter. Data was extracted from the binary storage files and analyzed using custom software written in C++ and Matlab (MathWorks, Natick, MA, all stimulus generation and analysis code is available for download at https://gavorniklab.bu.edu/supplemental-materials.html). Each animal was implanted with an electrode in both the left and right hemisphere. When electrodes produced a clear and comparable VEPs bilaterally, both hemisphere's responses were averaged together. Otherwise, we used data from the hemisphere with the largest VEP (in all cases, the same hemisphere was used for all recording sessions). VEPs were quantified by algorithmic scoring of the peak-to-peak voltage swing following a visual stimulus event. Sequence magnitudes are defined as the average peak-to-peak response magnitude for the second and third elements of the sequence, either B-C (trained) or C-B (novel).

## Stimulus delivery

Visual stimuli were generated with custom software written in Matlab using the PsychToolbox extension (http://psychtoolbox.org) to control stimulus rendering and timing. A 27-inch widescreen monitor (Acer XB270HU) was positioned 20 cm in front of the mouse and centered so as to occupy the entire binocular region of visual space. Visual stimuli consisted of full-field sinusoidal grating utilizing the full range of monitor display values between black and white, with gamma-correction to ensure constant total luminance in both gray-screen and patterned stimulus conditions. For acuity experiments, animals were exposed to 200 phase reversals at each spatial frequency. Phase reversals occurred every 0.5 secs. SRP experiments used a grating with spatial frequency of 0.05 cycles per degree. A single "familiar" orientation was presented 400 times on each training day, and a "novel" orientation was interleaved with the familiar stimulus on the test day. In Sequence Learning experiments, a sequence consisted of four elements of a full-screen, 100% contrast sinusoidal grating at 0.05 cycles per degree (each held on screen for 150 ms), followed by an inter-sequence gray period lasting 1.5 sec. Sequence elements differed by a minimum of 30 degrees and order was restricted to prevent the appearance of rotation. During training, a single sequence was presented 200 times per day in four blocks of 50 presentations with each block separated by 30 sec. On the test day, blocks of a novel sequence (DCBA) were interleaved with the trained (ABCD) sequence.

## Experimental design and statistics

All experiments comparing male and female mice were conducted using yoked littermates and blind to sex (though physical differences between male and female mice are sometimes apparent). Due to variations in stage onset and duration, it was not possible to fully yoke experimental groups in experiments addressing the effects of estrous cycling. These experiments were conducted on cage-mate animals, with staging and experiments occurring in parallel as much as possible. In all experiments, data was analyzed blind using a single common scoring algorithm.

All data are shown using population-averaged VEPs (e.g., the average stimulus-locked LFP) and violin plots of quantified VEP magnitudes. The shape of each violin indicates a kernel density estimates of the data (produced using the ksdensity function in Matlab's statistical toolbox) with mean values and data quartiles marked. To facilitate accurate comparisons, all violins on each individual plot were produced using a single bandwidth parameter chosen as the average of optimal values calculated for each individual data set on that plot. Statistical $n$ values reported indicate the number of individual animals in each experimental group. SPSS was used for parametric statistical analysis. Unless otherwise noted, 2-way ANOVAs were used to determine the statistical impact or either sex or estrus stage on stimulus evoked VEP

potentiation as a function of either training day or stimulus type and the Shapiro-Wilk test was used to confirm data normality. When main effects were significant, pairwise comparisons were performed using the independent two-tailed t-test with Bonferroni correction for multiple comparisons. The sizes of experimental cohorts were planned based on previously published experiments and our own experience indicating 5–10 animals are required in each cohort to reach statistical significance with adequate power. We used post-hoc estimates to verify that all statistically significant effects had an observed power $\geq 0.8$ for $\alpha = 0.05$ (true in all cases unless otherwise noted). In all cases, we planed the experiment using the minimum number of mice expected to be required to achieve significant results based on our expectation of attrition rates and effect sizes. Animals were excluded from the experiment only for electrode failure or as described in the text.

## Results

### Female and male mice have comparable visual acuity which is unaffected by estrous cycle

Our first set of experiments were designed to establish whether genomic differences during development result in any baseline functional differences in visual responsiveness between adult male and female mice. Visually evoked potentials (VEPs, calculated as the average stimulus-locked local field potential response) recorded in V1 can be used to assess visual function in mice and produce a quantitative metric that matches behavioral measures of visual acuity [35, 36]. Adult (P67) female and male littermate mice were implanted with VEP recording electrodes in layer 4 of binocular primary visual cortex, a depth that yields the maximum negative going VEP [33, 37]. Head-fixed animals were shown phase reversing sinusoidal gratings with 8 spatial frequencies between 0.05 and 0.7 cycles per degree (Fig 1A). As the spatial frequency increases, the magnitude of evoked potentials decreases [34] and the point at which the VEP asymptotes at the level of fluctuations recorded during gray-screen viewing identifies the upper limit of visual acuity (Fig 1B). While there is a clear and expected effect of stimulation spatial frequency on VEP magnitude ($F_{7,200} = 56.03$, $p < 0.001$), this metric revealed no statistical difference between the visual acuity of female ($n = 11$) and male ($n = 16$) mice ($F_{1,200} = 1.96$, $p = 0.16$) nor any interaction between sex and acuity ($F_{7,200} = 1.14$, $p = 0.34$).

We next repeated the acuity measurement in females to determine whether estrous cycle has a statistical effect on visual responsiveness, grouping female mice based on cytology of vaginal leukocyte and epithelial content (Fig 2). In our hands cycling occurred every 6–12 days in an irregular and unpredictable manner (Table 1), with often rapid progressions though metestrus and proestrus that would make it difficult if not impossible to design multi-day plasticity experiments reliably occurring during specific stages of the cycle across yoked cohorts. Visual acuity was assessed at the peak of estrus ($n = 9$) and diestrus ($n = 10$, Fig 1C). As with sex, there was a highly significant effect of spatial frequency ($F_{7,136} = 60.44$, $p < 0.001$) but measured no acuity differences between estrus and diestrus ($F_{1,136} = 0.564$, $p = 0.45$) and no significant interaction of the within and between animal factors ($F_{7,136} = 0.825$, $p = 0.57$).

### Sex influences some forms of experience dependent plasticity

Having established that there is no statistical difference in baseline visual physiologys of either sex or estrous stage, we next attempted to determine whether sex modulates experience-dependent cortical plasticity. Stimulus-selective response potentiation (SRP) is a form of visual learning induced by daily presentations of a visual stimulus of a particular orientation [28, 36, 38]. SRP is easily characterized by a significant potentiation of VEPs elicited by familiar stimuli

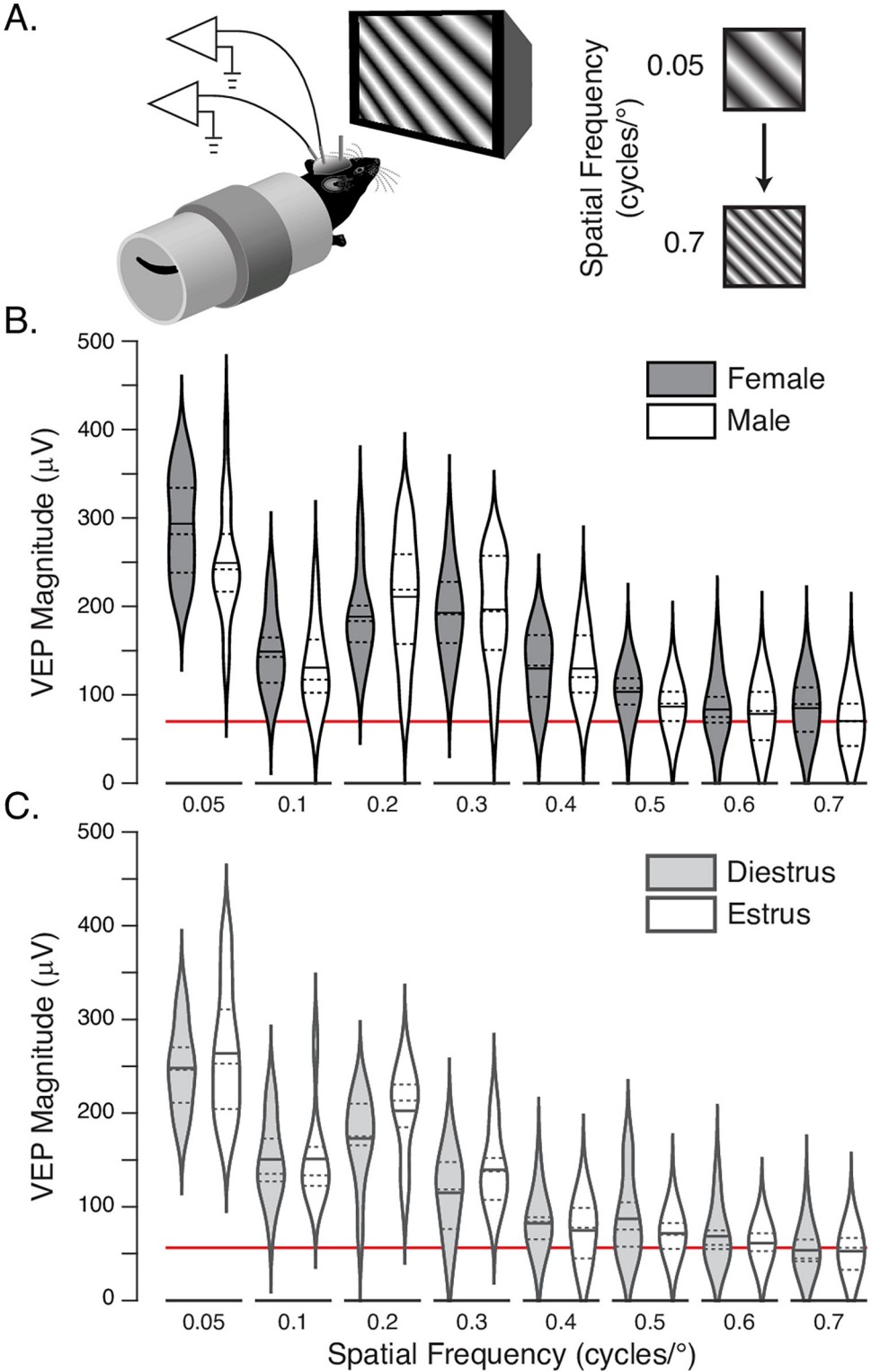

**Fig 1. Visual acuity as measured by VEPs is comparable in female and male mice. A.** Local field potentials are recorded from head-fixed mice (left) while they view phase-reversing sinusoidal gratings with spatial frequencies varying from 0.05–0.7 (right). **B**. Violin plots (dashed lines mark quartile boundaries and solid horizontal lines show the mean) showing the peak-to-peak magnitude of VEPs recorded in female (shaded) and male (white) mice as a function of spatial frequency. The red line marks the approximate noise-level recorded absent visual stimulation. **C.** The same as in B, but for female mice in either estrus (shaded) or diestrus (white). There is no effect of either sex or stage (see main text for statistical reporting).

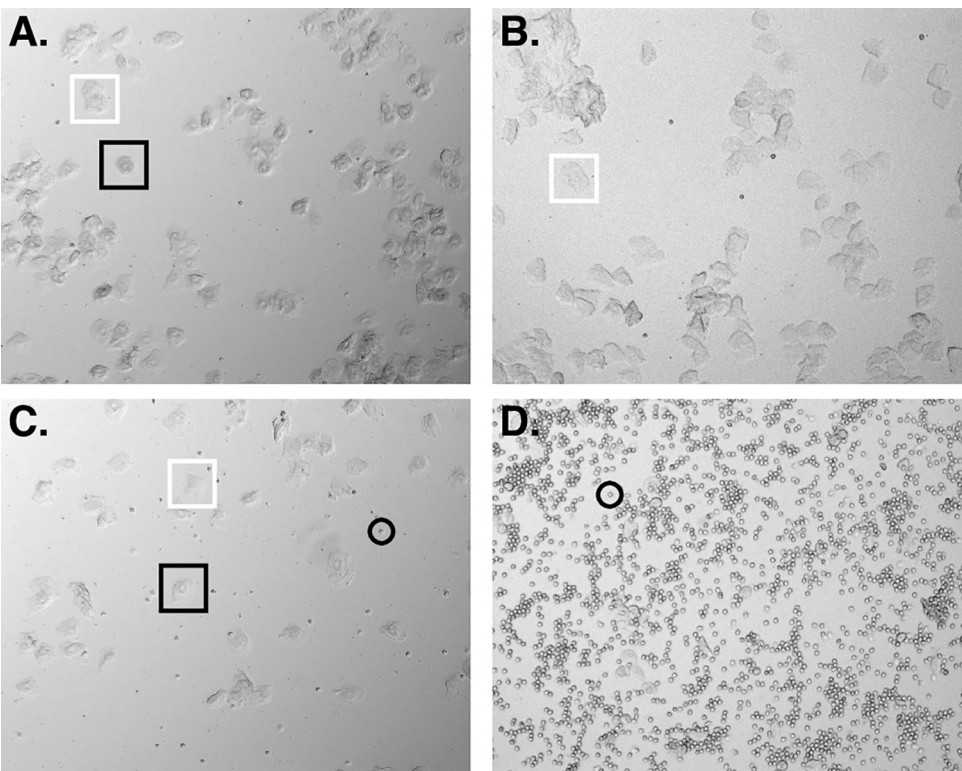

**Fig 2. Example images of estrous stages from unstained vaginal cytology.** Estrous cycle stage is determined by the presence of specific cell types. Nucleated epithelium, rounded with visible nucleus (*black boxes*). Cornified epithelium, flat, irregularly shaped with no visible nucleus (*white boxes*). Leukocytes, small and spherical (*black circles*). **A.** Proestrus, majority nucleated epithelium. **B.** Estrus, majority cornified epithelium. **C.** Metestrus, mix of cornified epithelium, nucleated epithelium, and leukocytes. **D.** Diestrus, majority leukocytes.

and provides a robust measure of underlying synaptic plasticity. This potentiation occurs over days and is selective for the spatial parameters of the stimulus used to induce it. SRP requires NMDAR signaling that results in AMPAR insertion at the synapse [28] and employs the mechanism of long-term synaptic potentiation (LTP) [38] including in parvalbumin-positive interneurons [39]. As mentioned above, these elements have been identified as potential mechanistic correlates of sex-dependent plasticity and might be expected to cause measurable plasticity differences between male and female mice.

Following the standard VEP implantation surgery, adult (approximately P67) littermate mice were presented with a phase-reversing 0.05 cy/° sinusoidal grating stimulus rotated to 45° every day for 5 days (Fig 3A). On the 5th day of the SRP experiment, the mice were also presented with a novel stimulus constructed by rotating the familiar stimulus to a new angle (135°). In accord with previous experiments, we found that VEP amplitudes evoked by the familiar visual stimulus increased significantly across presentation days in both male ($n = 16$) and female ($n = 11$) mice (Fig 3B and 3C, $F_{4,125} = 51.39$, $p<0.001$), but there was no main effect of sex ($F_{1,125} = 1.57$, $p = 0.21$) or significant interaction ($F_{4,125} = 1.622$, $p = 0.173$). To isolate the effects of potentiation, statistics were calculated using VEP score normalized relative to in-group average response magnitudes on day 1, though the same conclusions follow if statistics are calculated using raw VEP values instead (Day: $F_{4,125} = 48.84$, $p<0.001$; Sex: $F_{1,125} = 3.37$, $p = 0.07$; Sex*Day: $F_{4,125} = 0.81$, $p = 0.52$). VEPs evoked on day 5 by the novel stimulus (Fig 3C) were significantly smaller than those evoked by the familiar stimulus in both male

**Table 1. Estrous cycling in example animals.** Proestrus (*P*), estrus (*E*), metestrus (*M*), and diestrus (*D*).

| mouse | 1 | 2 | 3 | 4 | 5 | 6 | 7 | 8 | 9 | 10 | 11 | 12 | 13 | 14 | 15 | 16 | 17 | 18 | 19 | 20 | 21 | 22 | 23 | 24 | 25 | P | E | M | D |
|---|---|---|---|---|---|---|---|---|---|---|---|---|---|---|---|---|---|---|---|---|---|---|---|---|---|---|---|---|---|
| 1 | E | E | M | D | D | D | D | D | P | P | D | D | D | D | D | P | E | E | E | E | D | D | D | P | P | 5 | 6 | 1 | 13 |
| 2 | M | M | D | P | E | D | D | D | D | E | E | M | E | D | D | D | E | E | E | M | D | D | P | E | E | 2 | 9 | 4 | 10 |
| 3 | E | M | D | D | D | P | P | E | E | E | E | E | D | D | D | D | P | M | M | D | D | D | P | E | M | 4 | 7 | 4 | 10 |
| 4 | D | D | D | D | D | D | D | D | D | D | P | P | D | D | D | D | D | D | D | D | D | D | D | D | D | 2 | 0 | 0 | 23 |
| 5 | E | M | D | D | D | D | D | P | E | E | M | D | D | D | D | P | P | D | D | D | D | D | D | D | D | 3 | 3 | 2 | 17 |
| 6 | D | D | P | E | E | M | D | D | D | D | P | P | P | E | D | E | E | D | D | D | D | D | D | D | E | 4 | 6 | 1 | 14 |
| 7 | P | E | E | E | E | D | D | D | D | D | P | P | P | P | E | D | E | E | E | D | D | D | D | D | D | 5 | 8 | 0 | 12 |
| 8 | E | M | D | D | D | D | D | D | D | P | D | P | P | P | E | D | D | D | D | D | D | D | P | D | D | 5 | 2 | 1 | 17 |
| 9 | M | D | D | E | E | D | D | D | D | P | P | P | D | D | D | D | D | E | M | D | D | D | D | D | D | 3 | 3 | 2 | 17 |
| 10 | E | E | E | E | E | D | D | D | P | M | M | D | D | P | E | E | M | D | D | D | D | D | P | D | D | 3 | 7 | 3 | 12 |
| 11 | E | E | M | D | D | D | D | D | D | D | D | D | D | D | D | D | D | P | D | D | D | P | E | E | E | 2 | 5 | 1 | 17 |
| 12 | E | E | D | D | D | D | D | P | E | E | M | D | D | E | E | E | E | D | D | P | E | E | D | D | D | 2 | 10 | 1 | 12 |
| 13 | P | M | E | M | M | D | D | D | D | D | D | D | D | D | D | D | P | D | E | D | D | P | E | E | D | 3 | 4 | 3 | 15 |
| 14 | D | P | E | D | D | D | D | D | D | D | D | D | D | P | E | E | D | D | D | E | M | D | E | D | D | 2 | 5 | 1 | 17 |
| 15 | E | E | M | M | D | D | D | D | D | D | D | D | D | D | D | D | D | E | E | M | D | E | E | M | D | 0 | 6 | 4 | 15 |
| 16 | P | E | M | D | D | D | D | D | D | D | D | D | D | E | E | E | P | E | E | E | M | E | M | D | D | 2 | 8 | 3 | 12 |
| 17 | E | E | M | D | D | D | D | D | D | D | D | D | D | D | D | D | P | E | E | D | D | E | E | E | M | 1 | 7 | 2 | 15 |
| 18 | P | E | D | D | D | D | D | D | D | D | D | D | P | E | E | E | E | D | D | P | E | E | E | E | D | 3 | 9 | 0 | 13 |
| 19 | E | E | M | E | D | D | D | D | D | D | D | P | P | M | M | M | D | M | M | D | D | P | E | E | E | 3 | 6 | 6 | 10 |
| 20 | D | P | M | D | D | D | D | D | D | D | D | D | D | D | D | D | P | D | D | D | D | D | P | E | E | 3 | 2 | 1 | 19 |
| 21 | D | P | M | E | D | D | D | D | D | E | E | D | M | P | E | E | E | E | M | D | D | D | E | M | M | 2 | 8 | 5 | 10 |
| 22 | E | E | M | M | D | D | D | P | D | D | P | E | E | E | E | E | E | M | D | P | E | E | E | M | D | 3 | 11 | 4 | 7 |
| 23 | E | E | E | D | D | D | P | E | E | E | E | E | E | M | D | D | P | E | E | E | D | D | D | D | D | 2 | 12 | 1 | 10 |
| 24 | P | E | D | E | M | D | D | D | D | D | D | P | P | E | E | E | E | E | E | M | D | P | E | E | E | 4 | 11 | 2 | 8 |
| 25 | P | P | D | D | D | D | D | D | D | D | P | P | E | E | E | E | E | E | M | M | D | D | E | M | M | 4 | 7 | 4 | 10 |
| 26 | D | P | M | M | D | D | D | D | D | D | P | E | E | E | E | E | E | D | D | D | D | D | P | E | E | 3 | 8 | 2 | 12 |
| 27 | E | P | E | E | D | D | D | E | E | E | E | E | M | D | D | D | D | D | P | E | E | E | E | E | D | 2 | 13 | 1 | 9 |
| 28 | D | D | D | D | D | D | D | D | P | E | E | E | E | E | E | M | D | D | D | D | D | E | E | E | D | 1 | 9 | 1 | 14 |
| 29 | E | E | E | D | D | D | D | P | E | M | E | E | M | D | D | D | P | E | E | M | D | D | E | E | M | 2 | 10 | 4 | 9 |
| 30 | E | E | E | E | E | E | E | D | D | P | E | E | E | M | D | D | D | P | P | E | E | D | E | E | M | 3 | 14 | 2 | 6 |
| 31 | M | D | D | P | P | P | D | D | D | P | M | D | D | M | D | E | E | E | M | M | E | E | E | M | D | 4 | 6 | 6 | 9 |
| 32 | M | M | D | P | D | P | D | D | P | D | D | D | E | M | D | D | D | E | E | P | E | E | E | E | D | 4 | 7 | 3 | 11 |
| 33 | E | D | D | D | D | P | P | P | P | P | E | E | E | E | E | E | E | E | M | M | D | D | E | E | E | 5 | 12 | 2 | 6 |
| 34 | E | E | E | E | M | D | D | D | E | E | E | D | D | D | P | E | E | E | D | D | E | E | M | E | E | 1 | 14 | 2 | 8 |
| 35 | E | E | E | D | M | D | D | D | E | E | D | D | P | M | E | E | E | M | D | D | P | P | D | D | D | 3 | 8 | 3 | 11 |
| 36 | E | E | E | E | E | E | M | D | D | P | P | P | P | D | D | E | E | E | E | D | E | E | E | M | M | 4 | 13 | 3 | 5 |
| 37 | E | E | E | E | E | M | D | D | D | P | M | D | E | D | E | E | E | E | D | P | D | D | D | D | D | 2 | 10 | 2 | 11 |
| 38 | E | M | D | D | D | D | D | D | D | D | E | D | P | D | P | D | D | D | P | D | P | E | E | E | E | 4 | 6 | 1 | 14 |
| 39 | E | E | E | D | D | P | E | E | M | M | D | D | P | D | D | D | D | E | E | E | M | E | E | E | M | 2 | 11 | 4 | 8 |

*(Continued)*

**Table 1.** (Continued)

| mouse | 1 | 2 | 3 | 4 | 5 | 6 | 7 | 8 | 9 | 10 | 11 | 12 | 13 | 14 | 15 | 16 | 17 | 18 | 19 | 20 | 21 | 22 | 23 | 24 | 25 | P | E | M | D |
|---|---|---|---|---|---|---|---|---|---|---|---|---|---|---|---|---|---|---|---|---|---|---|---|---|---|---|---|---|---|
| | | | | | | | | | | | | | | | | | | | | | | | | | | | | | |
| 40 | M | P | E | M | E | E | E | E | M | E | E | E | E | D | M | E | E | E | E | E | M | M | D | D | E | 1 | 15 | 6 | 3 |
| 41 | E | D | D | P | P | P | E | E | D | D | D | D | E | E | M | D | D | D | D | D | D | D | D | P | E | 4 | 6 | 1 | 14 |
| 42 | E | M | D | D | E | E | E | E | D | D | E | E | M | E | D | D | D | E | E | M | D | D | P | E | E | 1 | 12 | 3 | 9 |
| 43 | E | E | D | D | E | E | E | M | D | P | D | E | M | D | P | P | M | M | E | E | E | M | D | D | E | 3 | 10 | 5 | 7 |
| 44 | E | E | M | D | E | E | E | E | D | D | E | E | E | M | D | D | E | E | E | E | D | D | E | E | E | 0 | 16 | 2 | 7 |
| | | | | | | | | | | | | | | | | | | | | | | | | | | | | | |
| 45 | D | D | D | D | P | E | E | E | E | E | E | E | E | D | D | D | D | P | E | E | E | M | E | E | E | 2 | 14 | 1 | 8 |
| 46 | E | E | E | E | D | P | D | E | D | D | D | D | E | E | E | E | E | M | M | E | M | D | D | D | E | 1 | 12 | 3 | 9 |
| 47 | E | D | P | P | D | P | E | E | E | E | E | E | E | D | P | E | D | D | E | E | E | D | D | D | E | 4 | 13 | 0 | 8 |
| 48 | E | M | D | D | D | P | E | E | E | E | E | E | E | E | M | E | E | E | E | E | M | M | D | D | E | 1 | 15 | 4 | 5 |
| 49 | P | E | D | D | D | E | E | E | E | E | M | D | E | E | E | E | E | E | E | E | M | M | D | P | E | 2 | 15 | 3 | 5 |

and female mice ($F_{1,50}$ = 93.19, $p<0.001$), there was a small but significant effect of sex ($F_{1,50}$ = 4.35, $p = 0.04$) and no significant interaction ($F_{1,50}$ = 0.365, $p = 0.55$). The effect of sex on day-5 familiar/novel comparisons was moderate and relatively low power with these group sizes ($\eta_p^2$ = 0.08, observed power = 0.53).

SRP encodes spatial features of a visual image, but V1 is also capable of encoding the spatio-temporal aspects of a visual sequence [30, 40, 41]. Like SRP, this learning causes the magnitude of visually evoked responses recorded in V1 to potentiate over days. Unlike SRP, this learning does not require NMDA receptors and can be prevented by antagonizing muscarinic acetylcholine receptors in V1 [30], and M2 receptors specifically [42]. Given that increasing estrogen levels have been shown to enhance signaling in cholinergic basal forebrain neurons which project robustly to V1 [43, 44], that estrogen can modulate the expression and function of mAChRs [45, 46], that M2 receptors have been implicated in estrogen-induced enhancement of hippocampal memory [47, 48], and that involvement of 7α-Estradiol can modulate ocular dominance plasticity in rat V1 [25] we reasoned that this form of learning might be more susceptible to sex differences than SRP.

To determine whether sex impacts sequence learning, we measured VEPs in female ($n = 15$) and male ($n = 12$) mice in response to 200 presentations of a sequence of four oriented sinusoidal gratings (ABCD, where each letter represents a unique orientation; Fig 3D) for five days. On the fifth day, both groups were shown the trained sequence and a novel sequence constructed by reordering the same elements (DCBA). Sequence evoked potentials increased with training in both male and female groups (Fig 3E) and were quantified (Fig 3F) by averaging the peak-to-peak responses of elements B and C (which show the largest potentiation and are not biased by the large response that occurs when a patterned stimulus follow the gray screen, as occurs in A and D). As in the SRP experiment, data was normalized by day 1 averages to isolate potentiation. As expected, the increase across days in this metric was highly significant ($F_{4,125}$ = 16.88, $p<0.001$). While responses in female mice potentiated less than they did in male mice (female: M = 1.86, SD = 0.84; male: M = 2.10, SD = 1.07) this was not a significant effect ($F_{1,125}$ = 3.09, $p = 0.08$) and there was no significant interaction ($F_{4,125}$ = 0.35, $p = 0.84$). Unlike in the SRP experiment, however, the interpretation of this data did change when statistics were calculated using raw voltage measurements (Fig 3G). In this case potentiation over days remained highly significant ($F_{4,125}$ = 16.51, $p<0.001$), and there was also a significant effect of sex ($F_{1,125}$ = 14.56, $p<0.001$), though still no significant interaction ($F_{4,125}$ =

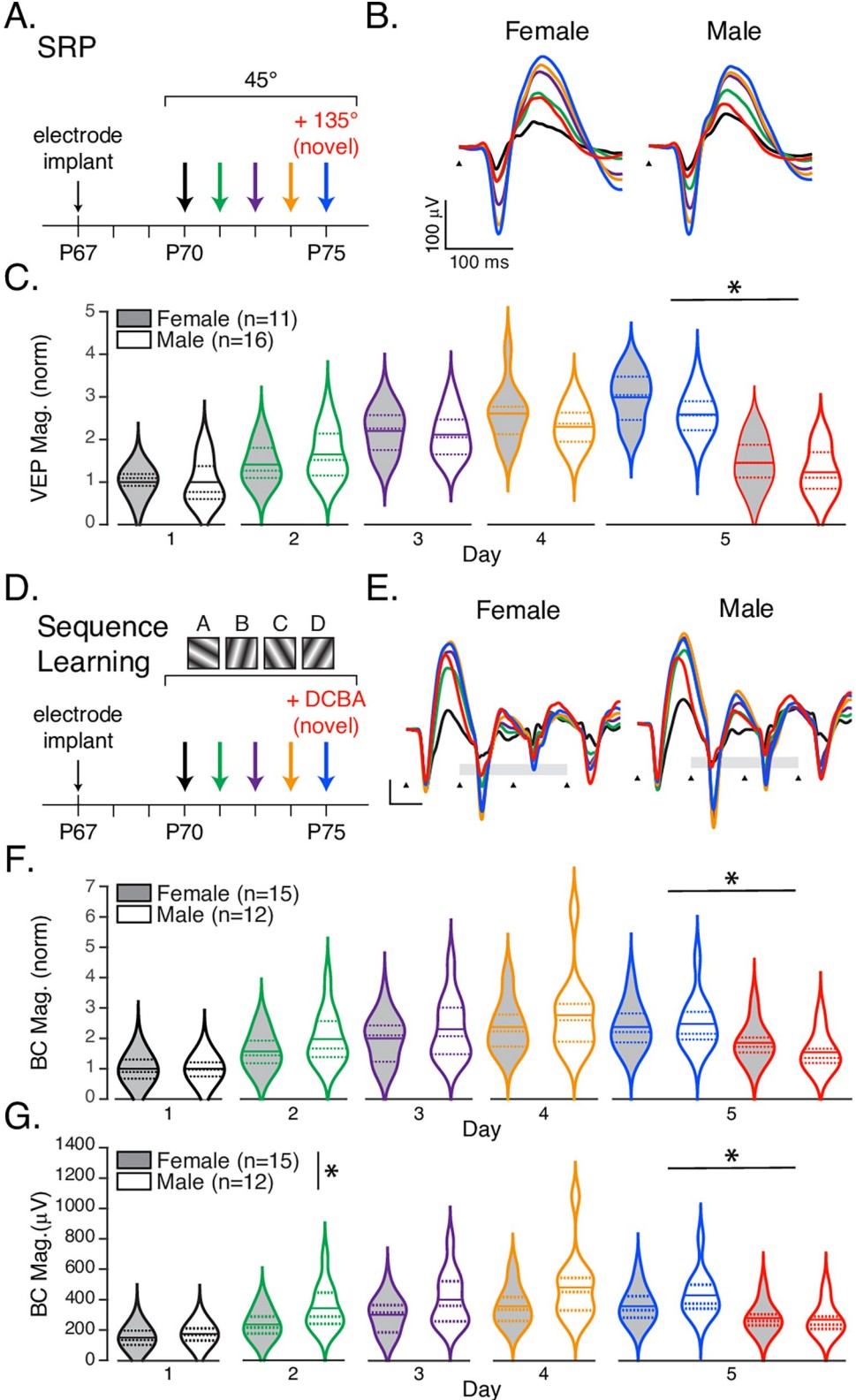

**Fig 3. Sex can affect experience dependent plasticity. A.** SRP experiment protocol. Animals were trained with repeated presentation of a 45° stimulus over five days. On the 5th day, a novel stimulus (red, 135°) was interleaved with the familiar stimulus (blue). **B.** Average VEP traces during SRP induction and expression in female and male mice,

color coded by day as in A. **C.** Violin plots showing the distribution of VEP magnitudes, normalized to group averages on day 1, of female (shaded) and male (white) mice as a function of experimental day. On day 5, novel responses are significantly smaller than familiar. **D.** In Sequence Learning experiments, mice were shown 200 presentations of the sequence ABCD every day for five days. On the fifth day, a novel sequence DCBA (*red*) was interleaved with ABCD (*blue*). **E.** Group averaged sequence responses during sequence learning (black triangles indicate sequence element onset times, the scale bar is 100 μV by 100 ms, color code as in D). **F.** Average quantified responses to elements B and C (indicated by the gray bars in E) of female (shaded) and male (white) animals normalized to day 1 group averages. **G.** The same as E but with raw voltages. The familiar sequence drives a significantly larger response than a novel sequence in both normalized and raw groups, but there is a significant effect of sex only in the raw data. See main text for detailed statistical reporting.

0.613 $p$ = 0.65). The statistical effect of sex was large ($\eta_p^2$ = 0.10, observed power = 0.97) and resulted from responses that were larger in male mice than females (female: M = 279.24, SD = 125.45; male: M = 364.98, SD = 187.28). On day 5, the trained sequence ABCD drove larger responses than did the novel sequence DCBA ($F_{1,50}$ = 14.41 $p$<0.001) though with no effect of sex ($F_{1,50}$ = 0.31, $p$ = 0.58) or interaction ($F_{1,50}$ = 1.13, $p$ = 0.29).

## Estrous stage affects experience dependent plasticity

There is extensive data showing that estrogen fluctuation impacts plasticity and learning in the hippocampus [49, 50] and prefrontal cortex [51], but there is little data on how sensory cortices are influenced by gonadal hormones [52]. We did not track or control for estrous cycle in the previous plasticity experiments. Since cycling occurs irregularly, it is possible that any effects of estrous phase on the induction or expression of plasticity averaged out across the population which could explain why there was no significant difference between the sexes in the 5-day SRP experiment. To address this possibility, we implanted mice at approximately P56 (female mice reach sexual maturity at approximately 8 weeks old) and began accumulating staging records after surgical recovery. The experimental approach was to compare the evolution of plasticity in groups starting at opposite ends of the estrous cycle, i.e. diestrus and estrus, so that the plasticity occurred under different hormonal conditions. To this end, female mice that had completed at least one full estrous cycle were sorted into yoked groups and exposed to the SRP induction protocol (Fig 4A). As before, normalized VEP magnitudes increased significantly over days (Fig 4B and 4C, $F_{4,115}$ = 20.10 $p$<0.001) and there was no interaction term ($F_{4,115}$ = 0.53, $p$ = 0.71), but VEPs in mice starting in diestrus potentiated ($n$ = 12, M = 1.71, SD = 0.0.32 on day 5) significantly more ($F_{1,115}$ = 5.404, $p$ = 0.02) than those starting in estrus ($n$ = 13, M = 1.51, SD = 0.20) with a small effect size ($\eta_p^2$ = 0.05, observed power = 0.64). On day five, the response to the novel stimulus is significantly smaller than to the familiar ($F_{1,46}$ = 2.67, $p$<0.001) with no significant effect of initial stage ($F_{1,46}$ = 2.08, $p$ = 0.16) or interaction ($F_{1,46}$ = 2.21, $p$<0.14). When the analysis is repeated using raw voltage values, the effect of the estrous cycle is more pronounced. On day 1, VEPs from mice in diestrus (M = 269.99, SD = 79.83) were smaller than those starting in estrus (M = 336.60, SD = 47.88) with highly significant effects of day ($F_{4,115}$ = 21.99, $p$<0.001) and stage ($F_{1,115}$ = 16.54, $p$<0.001), though still without a significant interaction ($F_{4,115}$ = 0.22, $p$ = 0.92). Comparing means between staging groups across days using Bonferroni corrected t-tests revealed that the difference between estrus and diestrus was significant on day 1 ($t$(23) = 2.05, $p$ = 0.04) and day 4 ($t$(23) = 2.50, $p$ = 0.01). There was no significant difference between stage cohorts on any other day.

Though all mice were grouped by estrous cycle stage on the first day of SRP induction, we found the cycling to be irregular with a high variability within each group across the 5 days of measurements (Table 1) which made it difficult to determine the extent to which the difference in estrous cycling averaged out in later days, and might explain the significant result on day 4 but not on days 2,3 or 5. In order to measure plasticity expression and induction within

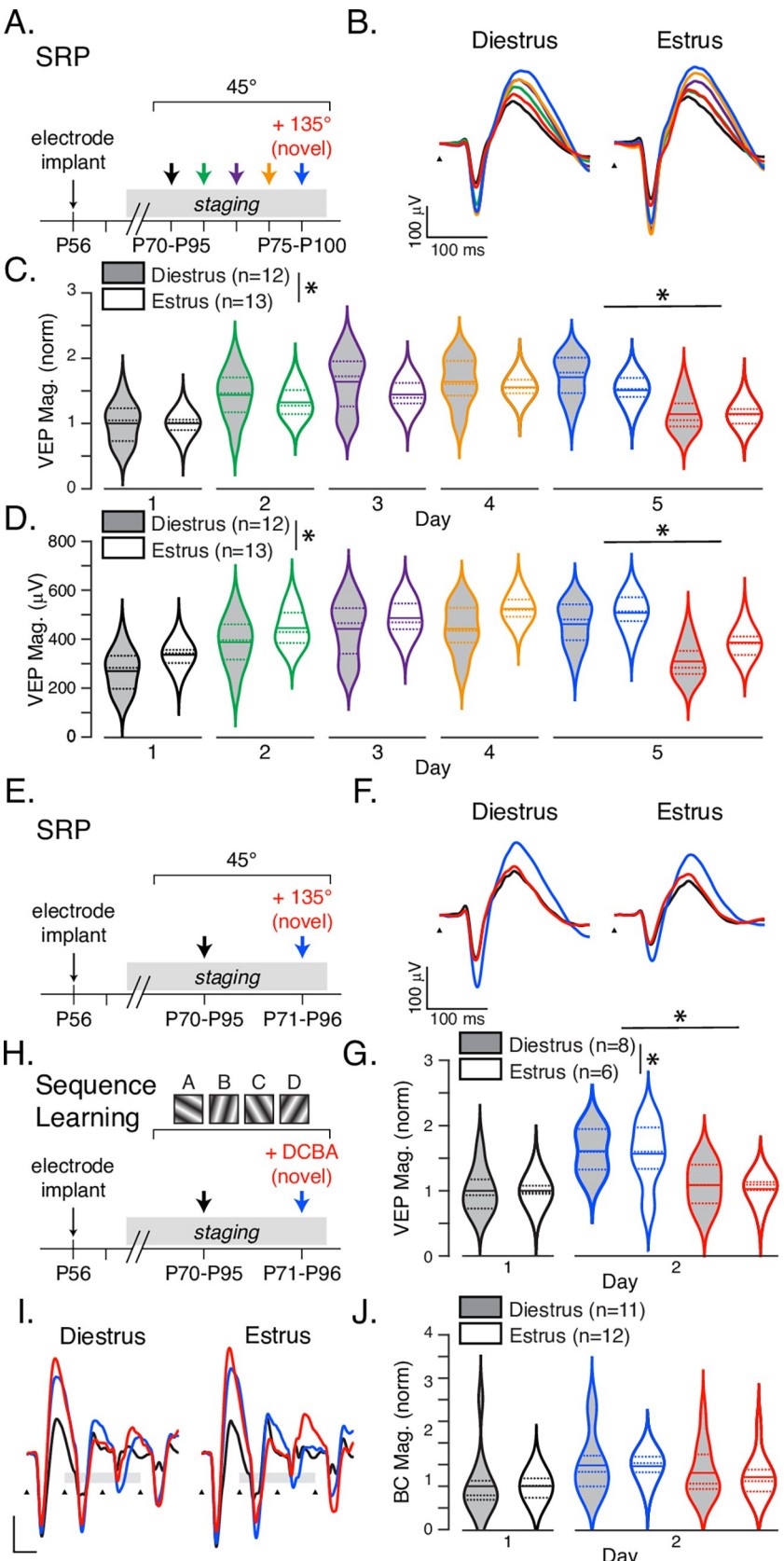

**Fig 4. Experience dependent cortical plasticity is affected by estrous stage. A**. Five-day SRP experiments were conducted with animals grouped by estrous cycle stage on day 1. **B**. Average traces over SRP induction and expression. Normalized, **C**., and raw, **D**., VEP magnitudes for mice in diestrus (gray) and estrus (white) stage on day 1. **E**. Two-day SRP experiments were conducted in animals that were in the same estrus stage on both days 1 and 2. **F**. Average VEPs and **G**. normalized quantifications or animals in diestrus or estrus during induction and expression. There was a significant effect of sex and familiar responses were significantly larger than novel on day 2. **H-J**. Two-day sequence learning experiment, average VEPs, and quantification as in E-G. See main text for statistical reporting.

specific hormonal windows, we repeated the SRP experiment using a single day of exposure with novel-stimulus testing occurring the following day (Fig 4E). Mice found to be in a different stage on day two (test day) relative to day one were removed from the data set (3 out of 17 mice were excluded for being in a different stage on day two, 2 diestrus animals entered estrus, 1 estrus mice entered diestrus). Analyzing normalized data (Fig 4G), we see that there is highly significant potentiation between days 1 and 2 ($F_{1,24}$ = 19.26, $p<0.001$). Diestrus ($n = 8$) mice potentiate slightly more than estrus ($n = 6$) mice after one day of SRP training ($F_{1,24}$ = 0.02, $p<0.001$) though this is a very small effect ($\eta_p^2$ = 0.001, observed power = 0.05). There is no significant interaction ($F_{1,24}$ = 0.02, $p$ = 0.88). On day two, there is no significant effect of stage ($F_{1,24}$ = 0.14, $p$ = 0.71) or interaction between stage and stimulus ($F_{1,24}$ = 0.01, $p$ = 0.94), but there is a significant difference between VEPs evoked by the familiar and novel stims ($F_{1,24}$ = 15.89, $p$ = 0.001). Analyzing raw voltages shows the same pattern, though the significant effect of stage between days 1 and 2 is noticeably larger ($F_{1,24}$ = 5.56, $p$ = 0.03, $\eta_p^2$ = 0.18, observed power = 0.62). Comparing Bonferroni corrected t-tests show that the difference between estrus and diestrus is significant only on day two (Day1: t(12) = 1.21, p = 0.24; Day 2: t(12) = 2.13, p = 0.04).

Having already found an effect of sequence between male and female mice, and owing to the difficulty of conducting an estrus-controlled 5-day experiment, we used the same abbreviated 2-day training protocol for the sequence stimulus (Fig 4H). Even with this narrowed 24-hour window, 7 out of 30 animals were excluded for being in a different stage on day 2 than day 1 (3 estrus animals entered metestrus, 2 diestrus entered proestrus, and 2 diestrus entered estrus). Analyzing normalized data showed that sequence responses potentiated significantly after one day of training ($F_{1,42}$ = 8.50, $p$ = 0.01), but surprisingly there was no significant difference ($F_{1,42}$ = 0.004, $p$ = 0.95) between mice in estrus ($n = 12$) and diestrus ($n = 11$) nor was there a significant interaction between stage and day ($F_{1,42}$ = 0.004 $p$ = 0.95). While the response to the novel sequence (M = 1.26, SD = 0.53) was smaller on average than to the familiar sequence (M = 1.47, SD = 0.54) after one day of training, this was not a significant effect ($F_{1,42}$ = 1.75 $p$ = 0.19) and there was no effect of stage ($F_{1,42}$ = 0.13 $p$ = 0.72) or interaction ($F_{1,42}$ = 0.06, $p$ = 0.81). Analyzing raw data produced the same result (Day: $F_{1,42}$ = 8.02, $p$ = 0.01; Stage: $F_{1,42}$ = 0.41, $p$ = 0.52; Stage*Day: $F_{1,42}$ = 0.03, $p$ = 0.86).

## Discussion

Many studies have addressed visual acuity in males and females, and their findings of specific physiological measures showing sexual dimorphism are often contradictory ([53–56]; but see also the lack of sex differences in [57]). There is a trend towards males having better acuity at high spatial frequencies [58, 59] and females having larger amplitude VEPs overall [55, 60, 61]. Rat data mirrors human findings in that female animals show larger VEPs [62], but this effect is limited to low spatial frequencies [63]. Further, estrogen signaling has been reported to modulate several forms of visual recognition and memory in females [20, 22, 64]. Our data does not reveal any statistically significant differences in visual responsiveness between male and female mice at either high or low spatial frequencies. We did see a trend towards female mice

having larger VEPs in response to low spatial frequencies (Fig 1A, 0.05 cycles/degree) in our first experiment, but in our SRP experiment VEPs in female mice started smaller than in males (Fig 2B, black line) and the relative magnitude of estrus/diestrus groups differed in our two staged SRP experiments (Fig 4B and 4F). These intra-cohort differences, which are fairly common, were within the standard errors of the data and underscore the importance of yoked treatment groups throughout the course of plasticity experiments.

Overall, our SRP findings suggest that sex and estrous cycle has only a small to modest effect on spatial learning requiring NMDARs. Our results show that sex and estrous cycle clearly affect sequence learning, however, which is not surprising given what we know about the mechanistic basis of this plasticity. Visual sequence learning requires muscarinic acetylcholine signaling [30, 42], and there is abundant literature revealing that estrogen modulates cholinergic activity in the rat brain: choline acetyltransferase (ChAT) mRNA levels fluctuate across the estrous cycle [65, 66] and estrogen administration increases ChAT mRNA [67], ChAT protein expression [44], acetylcholine release [68, 69], and choline reuptake at the synapse [70, 71]. Furthermore, estrogen attenuates the effects of scopolamine in passive avoidance, demonstrating functional muscarinic cholinergic receptors are required for a different across-day learning task [66]. This divergence between passive avoidance and visual sequence learning data may be attributable to differences in the impact of gonadal hormones on the hippocampus (which is required for passive avoidance [72]) and primary sensory cortex.

The hippocampus is one of the brain regions most dramatically influenced by the presence or absence of estrogen [73] and its presence is required for sequence potentiation [41], though the nature of this relationship is still under investigation. Sexual dimorphism has also been reported in the amygdala [74] and the prefrontal cortex [75, 76] and our results suggest potentially interesting parallels between sexual dimorphism in limbic system plasticity when compared to sensoricortical plasticity. This work also describes relatively simple plasticity assays in V1 that can potentially be used to probe the relation between sex hormones and functional plasticity *in vivo*.

How do our findings in the primary visual cortex relate to other cortical regions? It has been long recognized that cortical circuits are organized around a common architecture, leading to the hypothesis that all areas of the cortex implement a common set of algorithms [77, 78] and the notion that visual circuits can be understood as a proxy for the rest of the cortex [79]. Shared computational mechanisms support both short and long term memory in various cortical regions [80]. These observations imply that our findings in V1 will be relevant in other cortical areas. However, there is currently a lack of studies addressing sexual dimorphism in different sensory modalities and it's possible there are interesting areas of divergence. For example, an individual study in S1 has shown that nitric oxide signaling is necessary for male–but not female–whisker deprivation plasticity [19] suggesting that more complex experiments may reveal functionally relevant sex differences not yet explored in V1.

We assumed that our female mice would have 4–5 day cycles based on oft-stated rules for mice [9, 32] and designed our SRP experimental timeline around this window (Fig 4A). However, our staging data (Table 1) illustrates that description of an estrogen "cycle", with its implicit suggestion of predictable regularity, is something of a misnomer in lab mice. Cycling in our mice was both longer (6–12 days on average) and more unpredictable than we expected. For this reason, we amended our recording timeline to eliminate as much variability as possible when recording staged Sequence Learning and limited our protocol to 24 hours (Fig 4H).

Several factors might explain this variability. First, exposure to male mice increases cycle regularity and decreases length [81]. This "Whitten effect", however, occurs only through nearly direct contact with male urine or dirty bedding, and washing equipment between male and female mice is sufficient to avoid this confound. A second factor is age. A longitudinal

study of lifetime estrous cycling [82] detected regular 4–5 day cycles from 4 to 12 months of age, but cycles greater than 6 days were common for animals of 2–4 months (which includes our mice). The ability to combine housing of non-littermate females and reduce housing costs is a major advantage to including female mice, though this can effect estrous cycle length: individually housed females have 4–5 day regular cycles [32, 83], 3–5 group-housed females have ~8 day cycles with a distribution spread from 4 to 14 days [84], and 20 group-housed females often cease cycling altogether [84, 85].

Female mice have been excluded from experiments, in part, due to concern that daily staging would be necessary to track estrus. Fortunately, our findings suggest that the effects of this variability can probably be ameliorated through simple in-group normalization of male and female mice. The practical difficulties designing multi-day yoked experiments are non-trivial and should be carefully considered for any studies where estrous cycle is found to play a significant role on measured outcomes. While these considerations are known to experts in the field, our own experience and conversations with colleagues whose primary focus is not endocrinology suggest that many research labs underestimate the challenges associated with planning and executing experiments that track estrous cycle. Speaking to this group specifically: we found the methodology of [31] for daily staging referenced against the estrous cycle wheel graphic in Fig 1 of [32] to be effective. The process is fast (10 animals can be done in 15 minutes) and simple. The animals seemed unbothered by the process, and we noted neither signs of discomfort (e.g. squeaking) nor aggression (e.g., biting) during vaginal lavage. Other papers in the literature use various histological stains which require fixation or prolonged drying times [86, 87], but unstained leukocytes (Fig 2 *black circles*) and cornified epithelial cells (Fig 2 *white boxes*) are seeunambiguous, and nucleated epithelial cells (Fig 2 *black boxes*) are easily identifiable (for more unstained example images, see [88, 89]).

There are valid arguments for including both sexes in order to generate more complete database of primary research for use in developing medical treatments [90] and from those who plan to continue with male-only experiments [91]. While our data could potentially be used to support either position, by showing an effect of sex it clearly suggests limits on the general applicability of conclusions based on plasticity experiments which exclude female mice or subject them to gonadectomies. There is also practical value in including both sexes related to animal resource utilization, potentially reducing cost and waste. Overall, we think the minimal additional effort required to control for the additional variability associated with mixed-sex cohorts is worth in most V1 plasticity experiments, particularly given the NIH imperative to include sex as a biological variable.

## Acknowledgments

We thank Mike Baum for his advice in designing these experiments, and Lisa Stowers and Sandeepa Dey for their advice and technical support related to estrous staging. We are grateful to Elizabeth de Laittre for technical support.

## Author Contributions

**Conceptualization:** Rachel W. Schecter.

**Formal analysis:** Rachel W. Schecter, Cambria M. Jensen, Jeffrey P. Gavornik.

**Funding acquisition:** Jeffrey P. Gavornik.

**Investigation:** Rachel W. Schecter, Cambria M. Jensen.

**Methodology:** Jeffrey P. Gavornik.

**Supervision:** Jeffrey P. Gavornik.

**Visualization:** Rachel W. Schecter, Cambria M. Jensen, Jeffrey P. Gavornik.

**Writing – original draft:** Rachel W. Schecter.

**Writing – review & editing:** Rachel W. Schecter, Cambria M. Jensen, Jeffrey P. Gavornik.

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
