## [Decision Letter · Decision Letter 0]

12 Dec 2022

PONE-D-22-27303Sex and estrous cycle affect experience-dependent plasticity in mouse primary visual cortexPLOS ONE

Dear Dr. Gavornik,

Thank you for submitting your manuscript to PLOS ONE. After careful consideration, we feel that it has merit but does not fully meet PLOS ONE’s publication criteria as it currently stands. Therefore, we invite you to submit a revised version of the manuscript that addresses the points raised during the review process.

Please do follow useful directions to enhance the quality of your manuscript presentation. As soon as I received your response, I'll make final decision regarding your manuscript.

We look forward to receiving your revised manuscript.

Kind regards,

Dragan Hrncic

Academic Editor

PLOS ONE

Journal Requirements:

 “NIMH R00MH099654. JPG

NEI R01EY030200. JPG

Reviewers' comments:

Reviewer's Responses to Questions

**Comments to the Author**

1. Is the manuscript technically sound, and do the data support the conclusions?

Reviewer #1: Yes

Reviewer #2: Yes

2. Has the statistical analysis been performed appropriately and rigorously? 

Reviewer #1: Yes

Reviewer #2: Yes

3. Have the authors made all data underlying the findings in their manuscript fully available?

Reviewer #1: Yes

Reviewer #2: Yes

4. Is the manuscript presented in an intelligible fashion and written in standard English?

Reviewer #1: Yes

Reviewer #2: Yes

5. Review Comments to the Author

Reviewer #1: Rarely have I read a paper of such high quality. The writing is clear and concise, the data are analyzed with impeccable rigor, and the figure presentation is of the highest quality. The topic is of great importance, and the authors have done a very good job explaining the motivation for the study. They have set a high standard for the field. This will be a key reference text for all who study plasticity in the cerebral cortex.

Reviewer #2: Overall, I think that the study was well designed, well written, and informative for designing future studies to examine potential cellular or molecular differences between cortical plasticity in male and female rodents. I do have a few comments for improving the manuscript.

1. Figure 2 would be easier to understand if the images were labeled with the estrous phase being shown.

2. In Figures 3 and 4 there is no indication of which groups are different which makes it difficult for the reader to follow.

3. It would be helpful to see the sample sizes indicated in the different figures.

4. Was data analyzed be repeated measures ANOVA?

5. Starting at line 270, it indicates there are some sex differences but there is no reference to the figure showing the difference.

6. Why did the female and male groups not differ in the sequence learning? Was this due to sample size or did difference in estrous cycle phase obscure differences compared to the male group?

7. The introduction to the sequential learning proposes that the sequential learning may be more sensitive to sex differences than the SRP. Was the sequential learning more sensitive? The normalized data showed no difference, but the raw data showed a sex difference. Was the difference due to starting from a different baseline or from less potentiation? Was the ratio of the response on the fifth day to the response on the first day greater in the males?

8. Do you think that that differences detected are large enough to cause functional differences?

9. Are the amplitudes of the VEP affected by the placement of the electrode or the size of the brain?

6. PLOS authors have the option to publish the peer review history of their article (what does this mean?). If published, this will include your full peer review and any attached files.

Reviewer #1: No

Reviewer #2: No

---

## [Author Response · Author response to Decision Letter 0]

31 Jan 2023

Responses to Journal Requirements:

1. Our revied manuscript carefully follows the style guide.

2. This work was funded by two NIH grants, one from NIMH and one from the NEI. Neither institute had a role in study design, etc. nor did they pay salary. Here is the funding statement:

This work was funded by grants NIMH R00MH099654 and NEI R01EY030200. Neither institute took part in study design, data collection, or analysis or paid author salary.

3. Data not shown statements been removed from the paper. All data used for statistics, including both raw and normalized data, is available online via a link in the manuscript.

4. Our study did not use human subjects and needs no IRB approval or consent forms. The manuscript already includes details of our IACUC approved protocols.

5. Our reference list is correct.

Responses to Reviewer’s Comments:

Reviewer 1: We appreciate viewer 1’s enthusiastic support for this work. 

Reviewer 2:

1. Good suggestion, figure 2 has been modified.

2. The statistical details are included in the text, but at the reviewer’s suggestion we have also added indicators marking statistical difference to figures 3 and 4 and have also modified the figure legend to indicate significant differences.

3. Statistics, including n values, are reported in the text for all data. Following the reviewers comment we also added them to plot legends in figures 1, 3 and 4.

4. From the methods section: “Unless otherwise noted, 2-way ANOVAs were used to determine the statistical impact or either sex or estrus stage on stimulus evoked VEP potentiation as a function of either training day or stimulus type and the Shapiro-Wilk test was used to confirm data normality.”

5. For simplicity, in cases where normalized and raw data lead to the same statistical conclusions figure showed only the normalized plots. The average relative magnitudes of raw voltage traces can be seen on the voltage plots and all data (raw and normalized) are available for download from the repository cited in the paper. At the reviewer’s suggestion we added an explicit reference to figure 3C where significant day 5 differences are shown.

6. There is a small difference between male and female mice in sequence learning. The data shows that visual sequences drive slightly larger VEP responses in male populations than females, and that this difference accounts for the statistical effect found between the groups when calculated with raw (but not normalized) data. The details of this difference are included in the manuscript.

7. The difference between male and female mice found for raw voltages (see above) suggests sequence learning is more sensitive, but the 2-day induction experiment does not show increased sensitivity to estrous cycle. Overall the differences are small and can be controlled via normalization. The ratio of day 5 to day 1 was slightly larger in males (2.47x vs 2.38x) but this was not a large enough difference to make a statistical difference with normalized data.

8. Generally, no. The physiological differences are small enough that any functional dimorphism would likely be essentially de minimis. Confirming this with functional behavioral assays is beyond the scope of this work.

9. VEP morphology changes in a characteristic manner with electrode depth. Electrodes were placed at the same depth relative to the cortical surface in all mice which was confirmed via postmortem histology.

---

## [Decision Letter · Decision Letter 1]

14 Feb 2023

Sex and estrous cycle affect experience-dependent plasticity in mouse primary visual cortex

PONE-D-22-27303R1

Dear Dr. Gavornik,

We’re pleased to inform you that your manuscript has been judged scientifically suitable for publication and will be formally accepted for publication once it meets all outstanding technical requirements.

Kind regards,

Prof. Dr. Dragan Hrncic

Academic Editor

PLOS ONE

Additional Editor Comments (optional):

Reviewers' comments:

Reviewer's Responses to Questions

**Comments to the Author**

1. If the authors have adequately addressed your comments raised in a previous round of review and you feel that this manuscript is now acceptable for publication, you may indicate that here to bypass the “Comments to the Author” section, enter your conflict of interest statement in the “Confidential to Editor” section, and submit your "Accept" recommendation.

Reviewer #2: All comments have been addressed

2. Is the manuscript technically sound, and do the data support the conclusions?

Reviewer #2: Yes

3. Has the statistical analysis been performed appropriately and rigorously? 

Reviewer #2: Yes

4. Have the authors made all data underlying the findings in their manuscript fully available?

Reviewer #2: Yes

5. Is the manuscript presented in an intelligible fashion and written in standard English?

Reviewer #2: Yes

6. Review Comments to the Author

Reviewer #2: (No Response)

7. PLOS authors have the option to publish the peer review history of their article (what does this mean?). If published, this will include your full peer review and any attached files.

Reviewer #2: No

---

## [Editor Report · Acceptance letter]

20 Feb 2023

PONE-D-22-27303R1 

Sex and estrous cycle affect experience-dependent plasticity in mouse primary visual cortex 

Dear Dr. Gavornik:

I'm pleased to inform you that your manuscript has been deemed suitable for publication in PLOS ONE. Congratulations! Your manuscript is now with our production department. 

Kind regards, 

on behalf of

Professor Dragan Hrncic 

Academic Editor

PLOS ONE